# Validation of Fucoxanthin from Microalgae *Phaeodactylum tricornutum* for the Detection of Amyloid Burden in Transgenic Mouse Models of Alzheimer's Disease

A-Hyeon Lee [1], Sung-Chul Hong [2], Inwook Park [3], Soljee Yoon [3], YoungSoo Kim [3], Jinsik Kim [1,*] and Seung-Hoon Yang [1,*]

1. Department of Medical Biotechnology, College of Life Science and Biotechnology, Dongguk University, Seoul 04620, Korea; skandkgus@dgu.ac.kr
2. Natural Informatics Research Center, Korea Institute of Science and Technology (KIST), Gangneung 25451, Korea; schong@kist.re.kr
3. Department of Pharmacy, Department of Integrative Biotechnology & Translational Medicine, and Yonsei Institute of Pharmaceutical Sciences, Yonsei University, Incheon 21983, Korea; iwp95@alz.re.kr (I.P.); soljeeyoon@alz.re.kr (S.Y.); y.kim@yonsei.ac.kr (Y.K.)
* Correspondence: lookup2@dgu.ac.kr (J.K.); shyang@dongguk.edu (S.-H.Y.); Tel.: +82-31-961-5155 (J.K.); +82-31-961-5156 (S.-H.Y.); Fax: +82-31-961-5108 (J.K. & S.-H.Y.)

**Abstract:** The visualization of misfolded Aβ peptides by using fluorescence chemical dyes is very important in Alzheimer's disease (AD) diagnosis. Here, we describe the fluorescent substance, fucoxanthin, which detects Aβ aggregates in the brain of AD transgenic mouse models. We found that fucoxanthin from the microalgae *Phaeodactylum tricornutum* has fluorescent excitation and emission wavelengths without any interference for Aβ interaction. Thus, we applied it to monitor Aβ aggregation in AD transgenic mouse models. Aβ plaques were visualized using fucoxanthin in the brain tissue of APP/PS1 and 5×FAD mice by histological staining with different staining methods. By comparing fucoxanthin-positive and thioflavin S-positive stained regions in the brains, we found that they are colocalized and that fucoxanthin can detect Aβ aggregates. Our finding suggests that fucoxanthin from *P. tricornutum* can be a new Aβ fluorescent imaging reagent in AD diagnosis.

**Keywords:** fucoxanthin; Alzheimer's disease; amyloid plaque; fluorescent dye; AD diagnosis

## 1. Introduction

Alzheimer's disease (AD), the most common type of dementia, is characterized by the accumulation of abnormally folded amyloid-β (Aβ) proteins in the brain. Aβ monomers are produced from amyloid precursor protein (APP) via β- and γ-secretase activity [1]. Next, they begin to aggregate into oligomers, fibrils, and plaques. These misfolded Aβ proteins can cause neuronal degeneration in regions of the brain associated with memory and cognitive functions, eventually leading to symptomatic onset in patients with AD [1,2]. Therefore, confirming the presence of Aβ aggregation has become an important hallmark for AD diagnosis.

Several histological dyes are used for Aβ detection in postmortem brain sections. Thioflavin S (ThS) specifically binds to the β-sheet rich structure of Aβ aggregates, leading to increased fluorescence intensity of its emission spectrum, which enables the visualization of aggregated Aβ via fluorescence microscopy [3,4]. However, ThS detects other β-sheet rich protein complexes, including tau [5,6]; therefore, new dyes that specifically validate Aβ aggregate detection are required. Aβ-specific antibodies, such as 6E10 and A11, are used to visualize Aβ aggregates via immunohistochemistry; however, these require expensive and time-consuming methods. New histological staining dyes that use natural substances that easily label and visualize Aβ aggregates should be studied because they are cost effective and require a simple process.

Fucoxanthin is a carotenoid extracted from the microalgae *Phaeodactylum tricornutum* [7]. It reduces inflammation, decreases oxidative stress, and has an antiobesity effect, demonstrating its safety for use in animals and humans [8–11]. Moreover, fucoxanthin reduces Aβ-induced oxidative stress and proinflammatory cytokines in microglial cells and hippocampal neurons [12,13], inhibits Aβ aggregation, and alleviates cognitive impairment in vivo [14]. Additionally, the fluorescence spectrum of fucoxanthin has absorption bands at 448, 476, and 505 nm and emission bands at 630, 685, and 750 nm [15]. Therefore, we hypothesized that fucoxanthin will directly interact with Aβ aggregates and be a suitable fluorescent dye to detect Aβ accumulation in the brain.

Here, we report that fucoxanthin extracted from *P. tricornutum* can be used as a fluorescent dye targeting Aβ aggregates in different AD transgenic mouse models. Histological analysis was performed on the fixed mouse brain tissue of APP/PS1 and 5×FAD mice, which exhibit remarkable Aβ production and eventually excessive accumulation of Aβ in the brain. Brain tissue from aged mice was stained with different concentrations of fucoxanthin using two types of washing solutions. Fucoxanthin colocalized with ThS regardless of the type of washing solution, which demonstrated that it can detect Aβ via fluorescent visualization. This can verify the detection of Aβ with ThS.

## 2. Materials and Methods

### 2.1. Animals

Two types of transgenic mouse carrying mutations associated with AD, APP/PS1 (B6; C3-Tg(APPswe,PSEN1dE9) and 5×FAD (B6SJL-Tg(APPSwFlLon,PSEN1*M146L*L286V)) were used in this study. Both breeds were obtained from the Jackson Laboratory (Bar Harbor, ME, USA). Mice genotypes were confirmed via PCR analysis from tail biopsies following the protocols from Jackson Laboratory (Protocol no. 23370, 004462, 31769). All mice were housed in a laboratory animal breeding room at Dongguk University and maintained under a controlled temperature with 12–12 h light-dark phase. Food and water were available ad libitum. Brain tissue was collected from APP/PS1 and 5×FAD mice that were older than 6 months of age. All animal experiments were approved by the Institutional Animal Care and Use Committee at Dongguk University and performed in accordance with the regulation of institutional guidelines.

### 2.2. Preparation of Fucoxanthin from P. tricornutum

Fucoxanthin from *P. tricornutum* was prepared as described previously [10,16]. Freeze-dried *P. tricornutum* was supplied from Algaetech (Gangneung, Korea), and ethanol was purchased Daejung (Korea) for fucoxanthin extraction. Briefly, 100 g freeze-dried *P. tricornutum* (100 g) was sonicated with ethanol (1 L) at room temperature for 2 h. Solids were removed using Whatman No.1 filter paper (Thermo Fisher Scientific, New Zealand). Fucoxanthin was obtained by using a rotary evaporator at 35 °C.

### 2.3. Measurements of Fucoxanthin Fluorescent Properties

To obtain the excitation point, fucoxanthin stock in DMSO (40 mM) was diluted to 2 mM, 1 mM, 500 μM, and 250 μM. From each solution, 150 μL was loaded into a 96-well clear bottom plate (Nunc, Denmark), and absorbance was recorded at 2 nm increments from 230 nm to 850 nm. The peak through absorbance scan was specified as a possible excitation wavelength. Next, 150 μL of each solution was transferred to a 96-well black-bottom plate (Costar, WA, USA) to acquire an emission wavelength. By applying the excitation wavelength for fucoxanthin, the emission scans were measured in 2 nm increments up to 850 nm from the maximum absorption scan added 20 nm. The narrowest peak displayed through emission scans was determined as its emission point. Using the excitation and emission point, as obtained above, the fluorescence intensities of fucoxanthin when mixed with Aβ were scanned to monitor whether fucoxanthin shows a fluorescence shift in the presence of Aβ monomers or aggregates. Aβ (1–40) peptides in this study were synthesized by Fmoc solid phase peptide synthesis (SPPS) protocols as previously reported [17]. Aβ

monomers (20 μM) were prepared by diluting Aβ stock (10 mM) in DMSO stored at −80 °C. By incubating the same Aβ (20 μM) at 37 °C for 6 days, Aβ aggregates were also prepared. Fucoxanthin was diluted to the same concentration as above (2 mM, 1 mM, 500 μM, and 250 μM). Then, 25 μL of either Aβ monomers or Aβ aggregates (20 μM) and 75 μL of each fucoxanthin solutions were mixed in a 96-well black-bottom plate. The fluorescence spectrum of fucoxanthin in the presence or absence of Aβ was recorded at 2 nm increments at varying spectra range. Every fluorescent spectral scan was done under the microplate reader (Infinite 200 PRO, Tecan).

### 2.4. Histological Staining with Fucoxanthin and ThS

Extracted brain tissue was fixed in 4% paraformaldehyde (pH 7.4) and immersed in 30% sucrose for cryoprotection. The fixed brain samples were cut into 30 μm thick slices with a Cryostat Microtome (CM1860, Leica). For tissue staining, fucoxanthin stock in DMSO (40 mM) was diluted to 2 mM, 1 mM, 500 μM, and 250 μM in PBS and incubated on the sections for 1 h at room temperature. Next, sections were consecutively rinsed with 100, 90, and 50% ethanol and then PBS (organic solvent-based washing method), or three times with PBS only (water-based washing method). In addition, 500 μM of ThS (Sigma, St. Louis, MO, USA) dissolved in 50% ethanol was added to the same slides for 7 min at room temperature and then rinsed with 100, 90, and 50% ethanol and PBS successively (Figure 1).

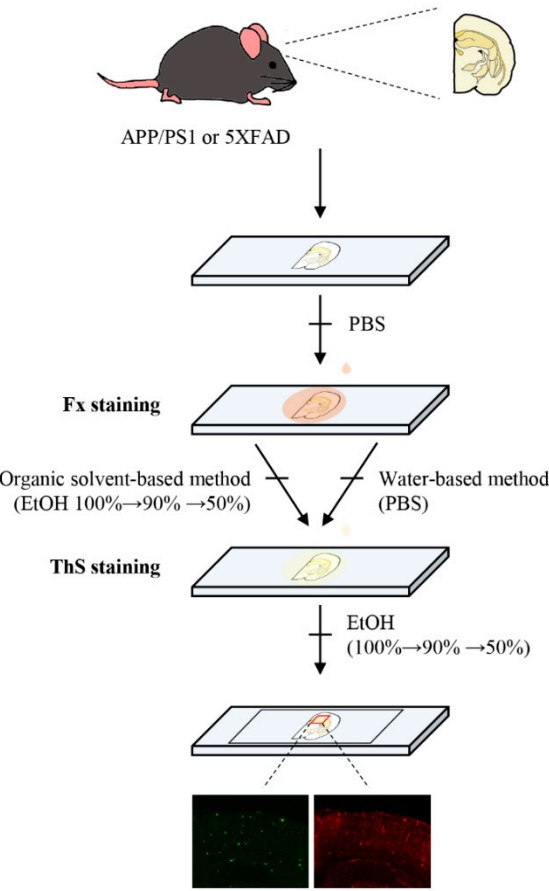

**Figure 1.** The histological staining process for fucoxanthin (Fx) in transgenic AD mice brain tissue. Different washing solutions, water- or organic solvent-based, were applied in the staining process.

## 3. Results

### 3.1. Fucoxanthin Has Fluorescent Properties Regardless of the Presence or Absence of Aβ Monomers or Aggregates

To specify the fluorescent spectrum of fucoxanthin extracted from microalgae *P. tricornutum*, we performed a fluorescent spectral scan of fucoxanthin at various concentrations (2 mM, 1 mM, 500 µM, and 250 µM). First, the absorbance scans were conducted to decide the excitation wavelength of fucoxanthin. Due to the high concentration, the absorbance exceeded the measurable range at 2 mM, but the absorbance decreased in a concentration-dependent manner at the concentrations of 1 mM, 500 µM, and 250 µM, and had maximum values at 430, 442, and 444 nm, respectively (Figure 2a).

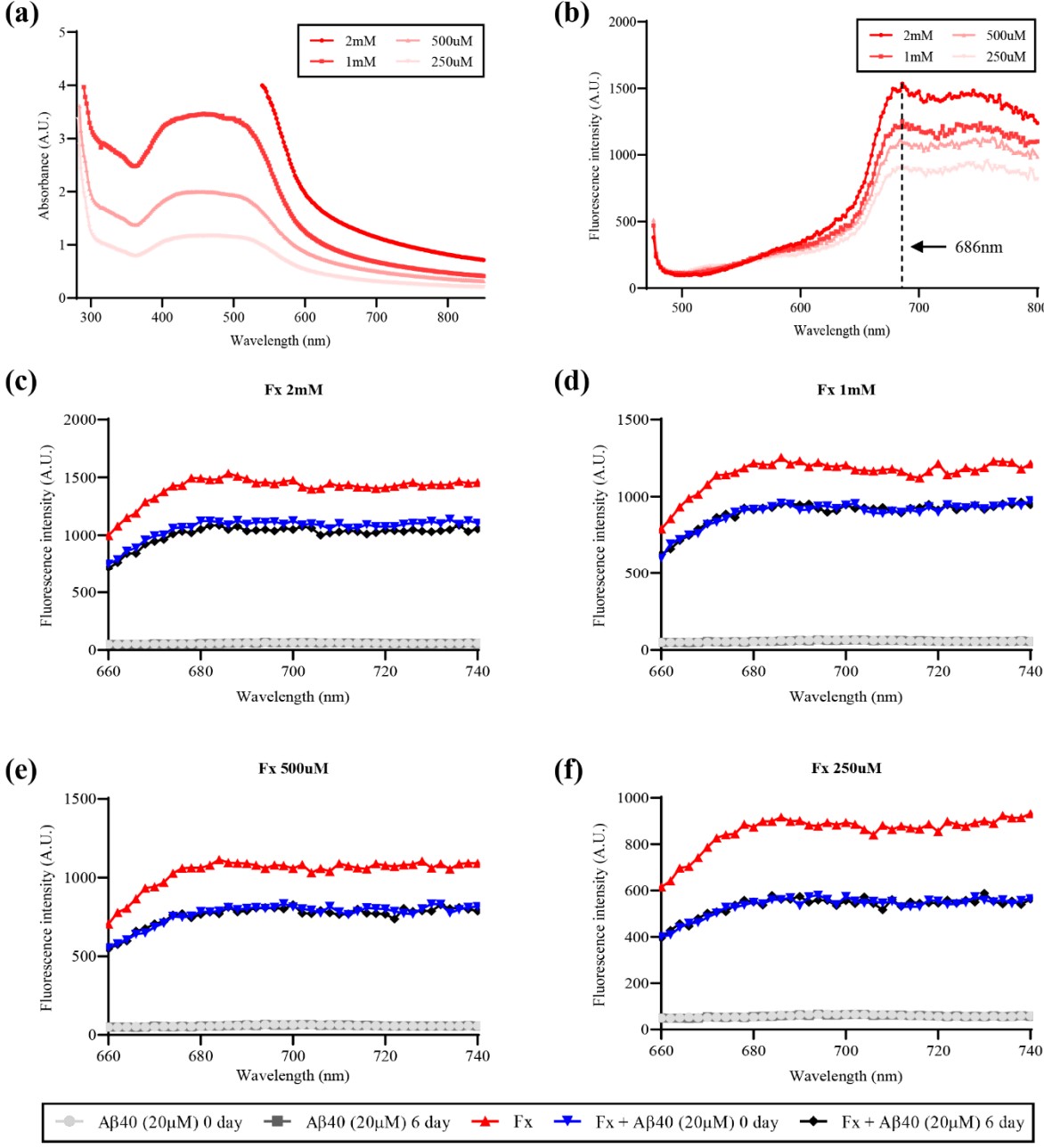

**Figure 2.** The fluorescent properties of fucoxanthin. Excitation and emission wavelengths of fucoxanthin at various concentrations were determined through absorbance (**a**) and emission scan (**b**); excitation, 444 nm; emission, 686 nm. Fluorescent spectra of each fucoxanthin solution with Aβ40 monomers or aggregates in a 3:1 ratio were recorded with excitation and emission wavelengths as above (**c**–**f**). A.U. = arbitrary unit.

Then, the emission scans of each fucoxanthin solution were measured using the excitation wavelength (444 nm) determined through an absorbance scan. In the emission spectrum, the fluorescence intensities of fucoxanthin reduced with decreasing concentration, and it had the narrowest peak at the same wavelength of 686 nm (Figure 2b). These results show that the fucoxanthin used in this study has fluorescent spectrum with an excitation point of 430–440 nm and an emission point of 686 nm, and thus can be monitored using a fluorescence microscope laser corresponding to the range.

It has been previously reported that fucoxanthin directly binds to Aβ peptide through hydrophobic interaction [14]. We conducted additional experiments to determine whether a fluorescence shift occurs when fucoxanthin binds to Aβ in brain tissue. Four concentrations of fucoxanthin, 2 mM, 1 mM, 500 µM, and 250 µM, were added to Aβ monomers (20 µM, 0-day incubation) or aggregates (20 µM, 6-day incubation) respectively. The emission scans of each mixed solution were measured by applying the wavelength as above. Consequently, fucoxanthin in the presence of Aβ monomers or aggregates showed a decrease when compared to the native fluorescence intensity of fucoxanthin itself (Figure 2c–f). However, it was observed that fucoxanthin has sufficient fluorescence intensity without losing its fluorescence properties, indicating that fucoxanthin could be applied as a fluorescence probe targeting Aβ.

### 3.2. Fucoxanthin Detected Amyloid Aggregates in AD Transgenic Mouse Models Using a Water-Based Washing Method

The popularly used AD transgenic mouse models, APP/PS1 and 5×FAD, possess mutations in APP and presenilin (PSEN1) genes, which are known to be associated with Aβ production. Thus, these mouse models show rapidly increased Aβ production and result in extensive plaque deposition and behavioral deficits [18]. As APP/PS1 and 5×FAD mouse models were originally generated only for the development of Aβ deposits, ThS single staining for Aβ plaque detection is widely used [19,20]. However, in the 5×FAD mouse model, it was found that ThS-positive tau aggregates make difficult the quantification of amyloid plaque. This suggests the necessity to develop new fluorescent dyes to support verification of Aβ plaque detection.

To investigate whether fucoxanthin can detect Aβ aggregates, we examined the histological staining in brain tissue from APP/PS1 and 5×FAD mice, with water- or organic solvent-based washing solutions during staining. We applied different concentrations of fucoxanthin in the brain for optimization during the staining process. Interaction between fucoxanthin and Aβ aggregates can be monitored by the comparison of ThS- and fucoxanthin-stained regions in the brains. As a result, we observed that 2 mM, 1 mM, 500 µM, and 250 µM of fucoxanthin was colocalized with ThS-positive staining in the hippocampus and cortex of both APP/PS1 and 5×FAD AD brain tissue when a water-based (PBS) washing solution was applied (Figures 3 and 4). Meanwhile, we confirmed that neither ThS nor fucoxanthin detected anything in wild-type mice (WT), where the AD phenotype is not observed (Figure S1 in the supplementary material). Our results indicate that fucoxanthin can detect Aβ aggregates in both APP/PS1 and 5×FAD mice with amyloid pathology, suggesting that fucoxanthin has an amyloid plaque specificity while excluding the possibility for ThS-positive tau aggregates observed in 5×FAD.

### 3.3. Fucoxanthin Staining Using an Organic Solvent-Based Washing Detected ThS-Stained Amyloid Aggregates in AD Transgenic Mouse

For reducing the nonspecific signal present in the brain sections during staining, the selection of washing buffer to eliminate fluorescent probes with lower binding affinity is an important consideration. Basically, the most frequently used buffer without destruction of cellular structure is phosphate-buffered saline (PBS). In this study, fucoxanthin from microalgae was extracted using ethanol, so this should take into account for the selection of buffer in the washing step. Thus, the Aβ staining ability of fucoxanthin needs to be evaluated by confirming its colocalization with ThS staining when using organic solvent-based washing solution in the fixed brain sections of AD transgenic mouse models. Similar

to the water-based washing solution application, Aβ plaques in APP/PS1 and 5×FAD brain tissues were stained with different concentrations of fucoxanthin (2 mM, 1 mM, 500 μM, and 250 μM) when an organic solvent-based (ethanol) washing solution was applied. These were colocalized with ThS staining (Figures 5 and 6). Taken together, these results suggest that fucoxanthin is a promising fluorescent imaging dye that can specifically detect amyloid aggregates in the AD transgenic mouse models regardless of the washing solution method.

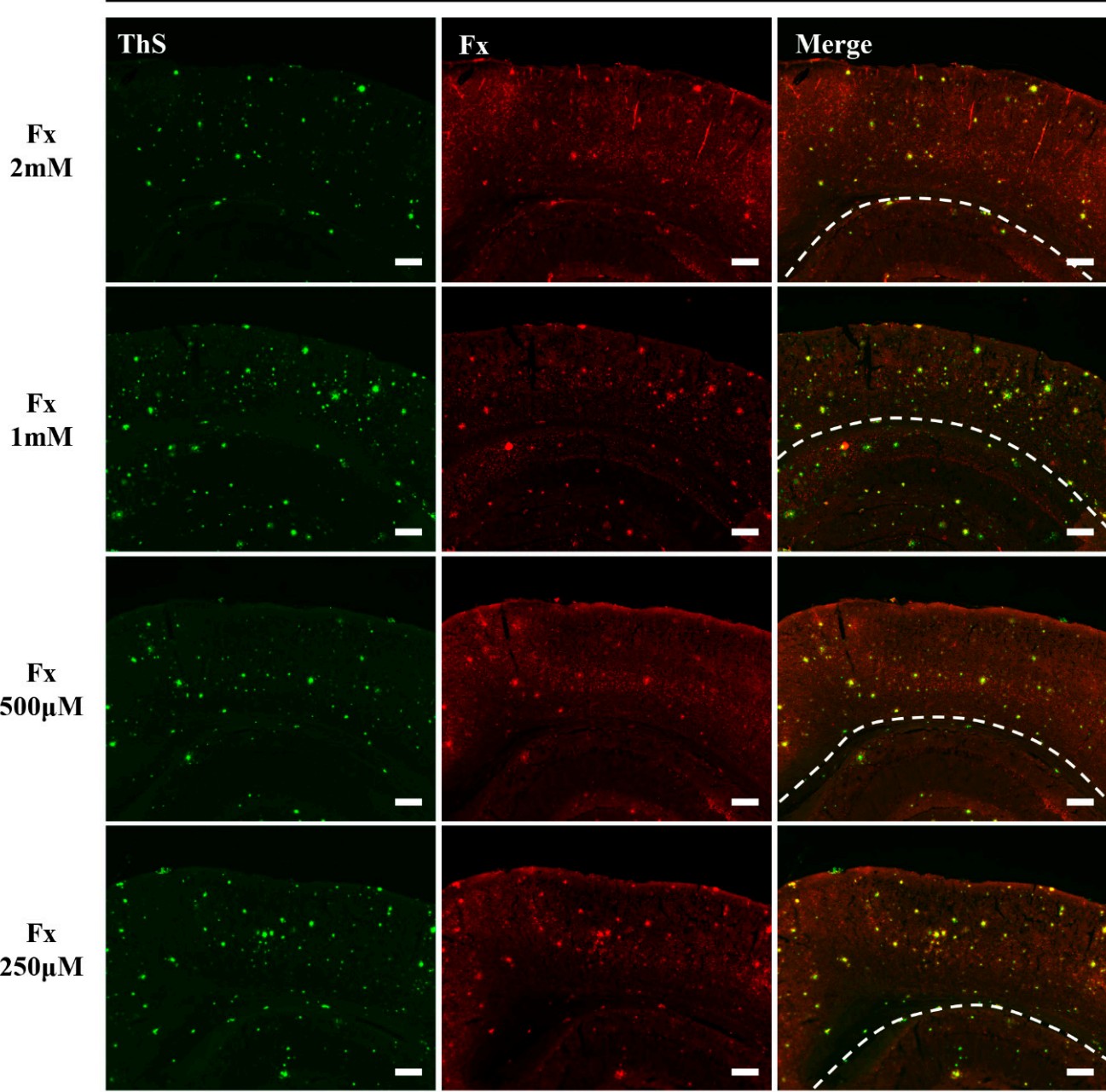

**Figure 3.** Fucoxanthin staining using the water-based washing method detected ThS-positive Aβ plaques in APP/PS1 brain tissue. The brains of 10–15-month-old APP/PS1 mice (*n* = 4) were visualized for Aβ plaques with fucoxanthin (Fx) and ThS. The images of ThS and Fx are merged to confirm that they are colocalized. Scale bars = 200 μm. The region of brain was separated with the white dash line on the images; upper region of dash line, cortex; under region of dash line, hippocampus.

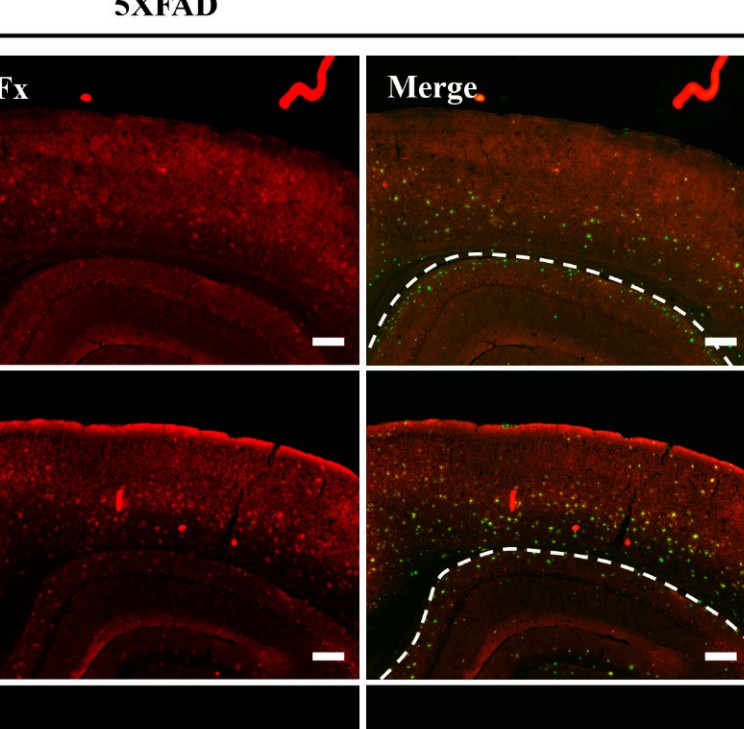
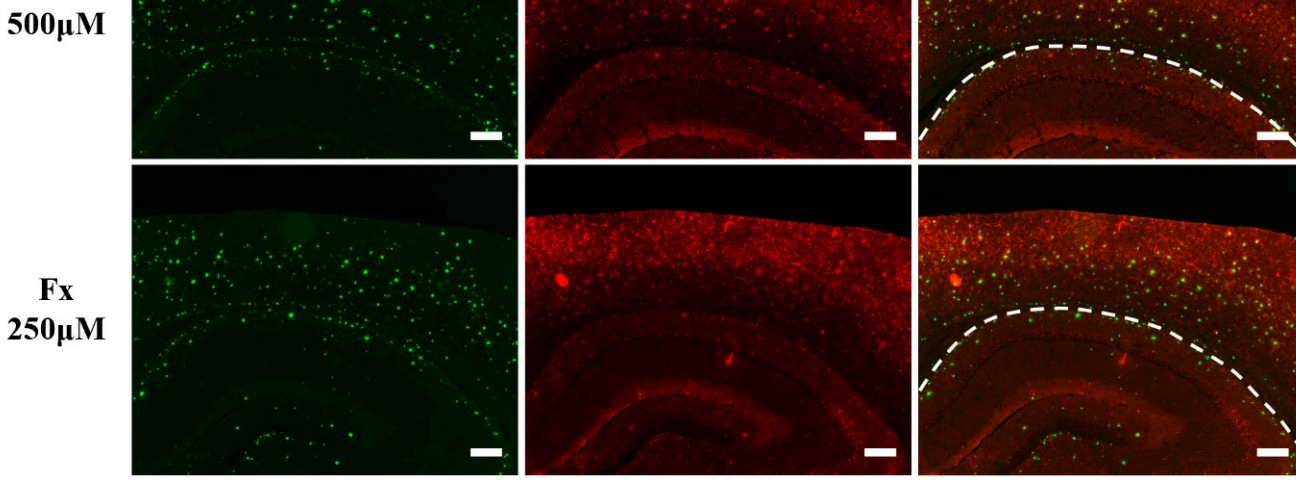

**Figure 4.** Fucoxanthin staining using the water-based washing method detected ThS-positive Aβ plaques in 5×FAD brain tissue. The brains of 10–15-month-old 5×FAD mice (*n* = 4) were visualized for Aβ plaques with fucoxanthin (Fx) and ThS. The images of ThS and Fx are merged to confirm that they are colocalized. Scale bars = 200 μm. The region of brain was separated with the white dash line on the images; upper region of dash line, cortex; under region of dash line, hippocampus.

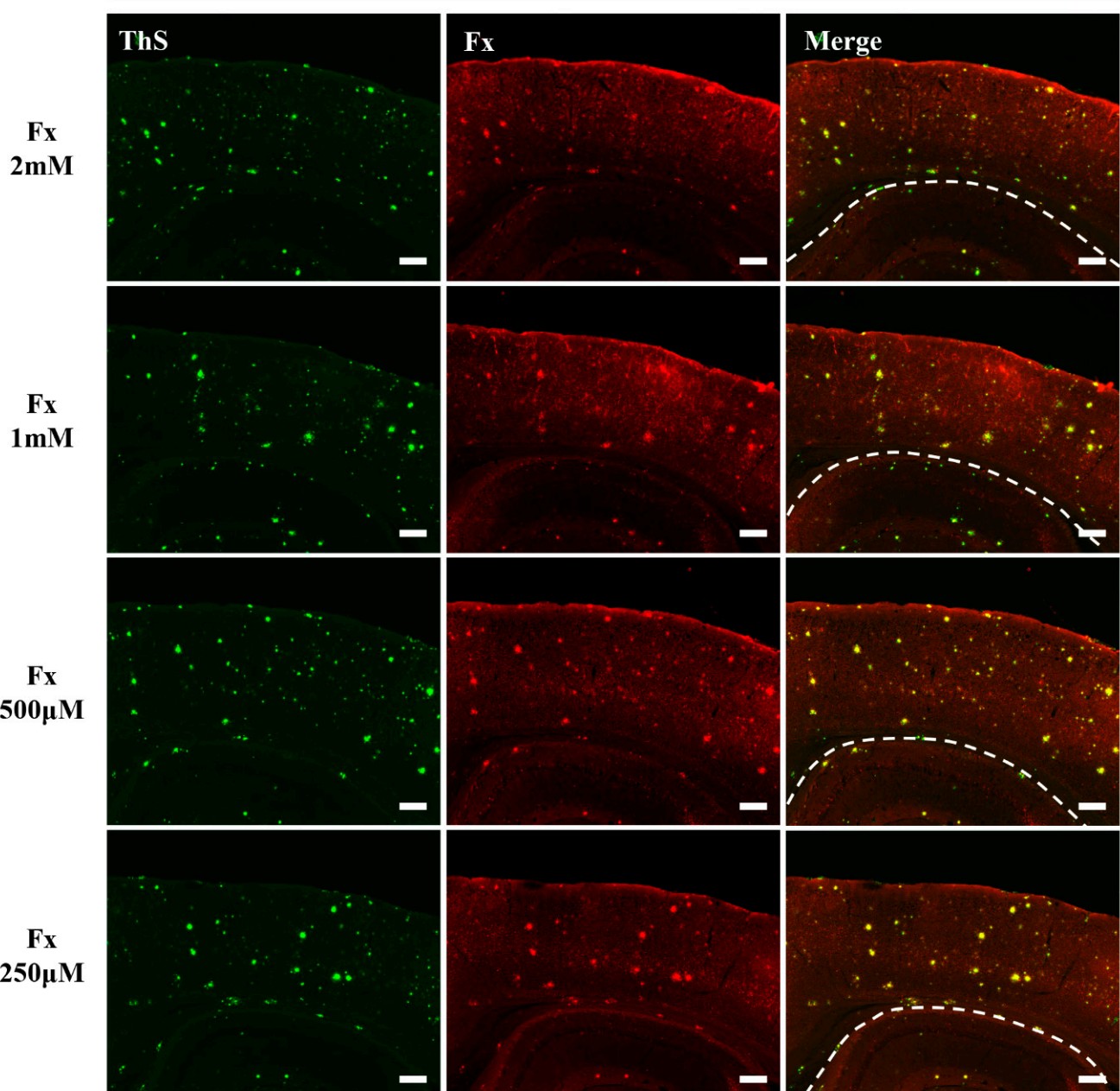

**Figure 5.** Fucoxanthin staining using organic solvent-based washing method detected ThS-positive Aβ plaques in APP/PS1 brain tissue. The brains of 10–15-month-old APP/PS1 mice (*n* = 4) were visualized for Aβ plaques with fucoxanthin (Fx) and ThS. The images of ThS and Fx are merged to confirm that they are colocalized. Scale bars = 200 μm. The region of brain was separated with the white dash line on the images; upper region of dash line, cortex; under region of dash line, hippocampus.

**5XFAD**

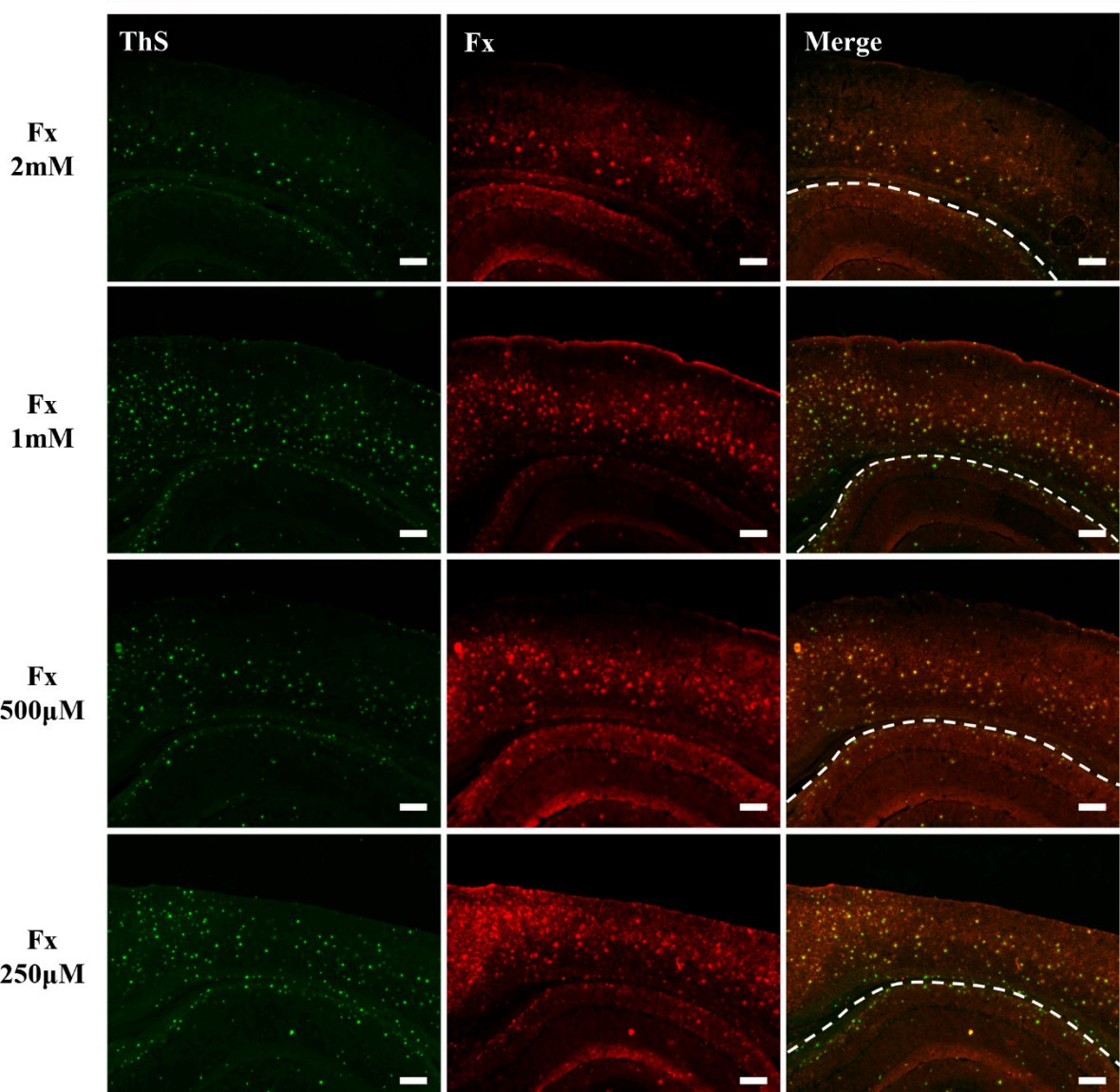

**Figure 6.** Fucoxanthin staining using organic solvent-based washing method detected ThS-positive Aβ plaques in 5×FAD brain tissue. The brains of 10–15-month-old 5×FAD mice (*n* = 4) were visualized for Aβ plaques with fucoxanthin (Fx) and ThS. The images of ThS and Fx are merged to confirm that they are colocalized. Scale bars = 200 μm. The region of brain was separated with the white dash line on the images; upper region of dash line, cortex; under region of dash line, hippocampus.

## 4. Discussion

Here, we report that fucoxanthin, extracted from the microalgae *P. tricornutum*, has a fluorescent spectrum with excitation and emission wavelengths (excitation: 430–440 nm, emission: 686 nm) regardless of the presence or absence of Aβ monomers or aggregates. When fucoxanthin was utilized as a fluorescent dye, it was colocalized with ThS-positive amyloid protein aggregates in the fixed brain tissue of AD transgenic mouse models, and different washing methods did not affect the histological staining of fucoxanthin.

Effective and accurate detection of Aβ, one of the pathological hallmarks of AD, is very important in its diagnosis and prognosis [21]. It is characterized by the cohesion of oligomers, fibrils, and plaques from monomers; therefore, specific detection substances for Aβ aggregates are used in AD diagnosis. ThS is most used to detect Aβ aggregates; however, recent studies have shown that ThS also detects tau protein aggregates in certain

AD transgenic mouse brains, such as 5×FAD [6]. Therefore, additional reagents capable of specific or supportable detection of Aβ aggregates is needed. The use of Aβ-specific antibodies enables very specific detection of different Aβ forms, but it takes at least 16–24 h [22,23]. Furthermore, the need for additional fluorescent secondary antibodies and chemicals significantly adds to the total cost for staining (Table 1).

**Table 1.** Comparison of fluorescent dyes for detecting Aβ in the brain section.

| Features | Thioflavin S | Aβ-Specific Antibody (6E10, A11, etc.) | Fucoxanthin |
|---|---|---|---|
| Duration | Less than 4 h | ~16–24 h | Less than 4 h |
| Additional chemicals | Ethanol | Fluorescent secondary antibodies, horse serum, PBS | Ethanol or PBS |
| Cost | ~$5/1 g | ~$200–300/one Aβ-specific antibody | Need to estimate |
| Specificity | Only fibrils containing β-sheet | Depending on the Aβ form, different specific antibodies are required | Need to further study |
| Stability | Stable in alcohol base solution | Depending on the secondary antibody conjugated with fluorescent dye | Stable in water base solution |
| Care after staining | Light sensitive | Light and temperature sensitive | Light sensitive |

Fucoxanthin is a carotenoid produced by plants and algae, including *P. tricornutum*, and it is already known to have beneficial biological properties such as anticancer, anti-inflammatory, and antiobesity physiological activities [10,12,16,24]. Recently, fucoxanthin has been reported to interact through binding to the amyloid protein, suggesting an association with AD [14]; therefore, the detection of Aβ aggregates via fucoxanthin could be presented as an additional imaging method that confirms Aβ detection via ThS staining. Like ThS staining, fucoxanthin is an advantageous staining procedure because it takes less than 4 h without additional chemicals. Furthermore, fucoxanthin can be mass-produced and has the potential to be applied to living animals as it is a naturally derived carotenoid, but ThS, which is a chemical material, is difficult to apply to clinical trials or AD patients since it is not proven yet. In contrast, we confirmed that fucoxanthin stained with oligomeric tau in the brains of 5×FAD, not in the WT and APP/PS1 brains (Figures S2–S4). Although fucoxanthin may not overcome all shortcomings of other dyes, it is expected that it will help to further refine AD diagnosis by reducing false-positive misdiagnosis as much as possible by at least supplementing the dyes.

A precise analysis of whether fucoxanthin interferes with other hallmarks of AD, such as tau aggregation, is required. Additional studies should investigate whether fluorescence imaging with fucoxanthin can be translated into diagnosis and prognosis in AD.

**Supplementary Materials:** The following are available online at https://www.mdpi.com/article/10.3390/app11135878/s1, Figure S1: Histochemical analysis of the brains in wild-type (WT) mice after staining with fucoxanthin (Fx) and thioflavin S (ThS), Figure S2: Histochemical analysis of the brains in wild-type (WT) mice after staining with fucoxanthin (Fx) and anti-oligomeric tau antibody, Figure S3: Histochemical analysis of the brains in 5XFAD mice after staining with fucoxanthin (Fx) and anti-oligomeric tau antibody. Figure S4: Histochemical analysis of the brains in APP-PS1 mice after staining with fucoxanthin (Fx) and anti-oligomeric tau antibody.

**Author Contributions:** Conceptualization, A.-H.L., S.-H.Y.; investigation and data curation, A.-H.L., S.-C.H., S.-H.Y.; resources, I.P., S.Y., Y.K.; Writing—original draft, A.-H.L., S.-H.Y.; Writing—review and editing, A.-H.L., J.K., S.-H.Y.; Funding acquisition, J.K., S.-H.Y. All authors have read and agreed to the published version of the manuscript.



**Funding:** This work was funded by the Dongguk University Research Fund of 2019 (S-2019-G0001-00009), National Research Foundation of Korea (NRF-2020R1F1A1076063), and the Korea Institute of Industrial Technology (Project no. EO170047).

**Institutional Review Board Statement:** The study was conducted according to the guidelines of the Declaration of Helsinki, and approved by the Institutional Animal Care and Use Committee at Dongguk University.

**Data Availability Statement:** Not applicable.

**Conflicts of Interest:** The authors declare no conflict of interest.

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
