# Peer review of "Validation of Fucoxanthin from Microalgae Phaeodactylum tricornutum for the Detection of Amyloid Burden in Transgenic Mouse Models of Alzheimer’s Disease"

_applsci, doi:10.3390/app11135878_

Round 1

Reviewer 1 Report

Lee et al. described in presented work fucoxanthin isolated from microalgae as dye for amyloid beta plaques fluorescence detection in AD transgenic mouse models. My question is, how can be fucoxanthin use for detection of amyloid beta aggregation, when there are evidences that fucoxanthin inhibits amyloid beta assembly and reduces amyloid beta oligomer-induced cognitive impairments (DOI: 10.1021/acs.jafc.7b00805, DOI: 10.1155/2017/6792543)??? The influence and pros and cons of fucoxanthin using on amyloid beta aggregation are suggested to address more clearly!

Author Response

Reviewer #1:

First of all, we would like to express our genuine gratitude for the time and effort put into reviewing our manuscript and providing comments that make this even better. All authors have carefully reviewed the comments in detail and revised the manuscript with care.

Lee et al. described in presented work fucoxanthin isolated from microalgae as dye for amyloid beta plaques fluorescence detection in AD transgenic mouse models. My question is, how can be fucoxanthin use for detection of amyloid beta aggregation, when there are evidences that fucoxanthin inhibits amyloid beta assembly and reduces amyloid beta oligomer-induced cognitive impairments (DOI: 10.1021/acs.jafc.7b00805, DOI: 10.1155/2017/6792543)??? The influence and pros and cons of fucoxanthin using on amyloid beta aggregation are suggested to address more clearly!

We really appreciate your in-depth comment on the influence of fucoxanthin when detecting amyloid plaques. As you mentioned, it has already been shown that fucoxanthin inhibits Ab assembly. However, when amyloid plaques are formed in AD, it is important not only inhibiting the aggregation of Ab but also disassociating the pre-aggregated plaques. Furthermore, it is not defined whether fucoxanthin can disassemble the plaques which have already formed. Moreover, through the reference paper (DOI: 10.1021/acs.jafc.7b00805), we realized that fucoxanthin can directly bind to Ab thus we hypothesized that fucoxanthin can also bind to Ab that has already been aggregated or formed plaques. Following experiment should be noted that further studies are needed, but we aim to determine that fucoxanthin may utilize as a tool to detect Ab plaques structurally rather than functional role in amyloid plaques. Since reviewer #2 also gave similar comments to yours on the pros and cons of fucoxanthin, additional explanations have been added to the discussion section of the manuscript to make it clearer (Line 221-224 of Page 10). Thank you again for your impressive comments.

Reviewer 2 Report

In the present paper, the authors have investigated the issue of visualization of misfolded Aβ peptides by using fluorescence chemical dyes. This issue is very important in the diagnosis Alzheimer’s disease (AD) patients. They describe the fluorescent substance, fucoxanthin from the microalgae, Phaeodactylum tricornutum, which detects Aβ aggregates in the brain of AD transgenic mouse models. Where they used different staining methods and we found that fucoxanthin can detect Aβ aggregates and as such propose it as new Aβ fluorescent imaging reagent in the diagnosis of AD. 
The paper has some flaws that need to be corrected and proofread by native speaker. 
Here are some comments:

In all figure legends the authors should describe which section of the brain is shown. How many experiments (from how many animals) were performed. Denote layers, cells... .....

The authors must perform additional experiments to show whether fucoxanthin labels tau protein and to what extent, so that they could argue that labelling with fucoxanthin is superior than other methods.

The authors should elaborate the discussion a bit more. For example, they already state that "A precise analysis of whether fucoxanthin interferes with other hallmarks of AD, such as tau 222 aggregation is required." This experiment is really important. And authors should convince the reader how fucoxanthin staining is better than ThS-labeling. It appears that fucoxanthin labels a significant amount of the tissue. What other structures are also labelled by fucoxanthin?

Absolutely important would be the experiment where the authors would show labelling in the control WT (no AD) animal to see what fucoxanthin labels in a healthy brain.

Without these additional experiments I cannot agree with publication of the paper.

Author Response

Reviewer #2:

First of all, we would like to express our genuine gratitude for the time and effort put into reviewing our manuscript and providing comments that make this even better. All authors have carefully reviewed the comments in detail and revised the manuscript with care.

In the present paper, the authors have investigated the issue of visualization of misfolded Aβ peptides by using fluorescence chemical dyes. This issue is very important in the diagnosis Alzheimer’s disease (AD) patients. They describe the fluorescent substance, fucoxanthin from the microalgae, Phaeodactylum tricornutum, which detects Aβ aggregates in the brain of AD transgenic mouse models. Where they used different staining methods and we found that fucoxanthin can detect Aβ aggregates and as such propose it as new Aβ fluorescent imaging reagent in the diagnosis of AD. 
The paper has some flaws that need to be corrected and proofread by native speaker. 
Here are some comments:

In all figure legends the authors should describe which section of the brain is shown. How many experiments (from how many animals) were performed. Denote layers, cells...

We apologize for our insufficient explanation regarding the conditions of the experiment in the figures. The areas of the brain shown in the results of the entire experiment we performed are the cortex and hippocampus, which are known to be the places where amyloid plaques accumulate. We marked dash lines in the merged fluorescence images to distinguish between the cortex and the hippocampus, detailed descriptions are given in the figure legend.

The authors must perform additional experiments to show whether fucoxanthin labels tau protein and to what extent, so that they could argue that labelling with fucoxanthin is superior than other methods.

The authors should elaborate the discussion a bit more. For example, they already state that "A precise analysis of whether fucoxanthin interferes with other hallmarks of AD, such as tau 222 aggregation is required." This experiment is really important. And authors should convince the reader how fucoxanthin staining is better than ThS-labeling. It appears that fucoxanthin labels a significant amount of the tissue. What other structures are also labelled by fucoxanthin?

We sincerely appreciate your valuable comments. We also fully agree with your comments that further experiments are needed to see if tau can be detected with fucoxanthin to argue that fucoxanthin staining is a better way. However, it is known that APP/PS1 and 5XFAD mice, which are AD transgenic mouse models used in our experiments, do not show the tau phenotype [1-3]. Although it was recently reported that tau pathology was observed in 5XFAD mice, we thought that this should increase reliability through further validation. Based on your insightful opinion, we performed immunostaining of oligomeric tau with fucoxanthin in two types of AD transgenic mice we used. As a result, as already known, oligomeric tau was not detected in APP/PS1 mice, and it was confirmed that oligomeric tau and fucoxanthin were co-labeled in 5xFAD mice. But, as mentioned earlier, we thought that more clear validation is needed for the tau phenotype in 5XFAD mice, so we focused on the ability of fucoxanthin to detect Ab. These results have been added in the discussion part and supplementary information.

Additionally, a notable advantage of fucoxanthin compared to ThS is that it is derived from natural sources. Advantages of fucoxanthin as a natural product may include: 1) Unlike ThS, which is a chemical compound, fucoxanthin can be mass-produced from natural materials such as microalgae, so it can effectively lower unit cost. This is also a strong advantage when compared to antibodies. 2) As can be seen from the results shown in our manuscript, fucoxanthin can be stained using both organic- or water-based solvents. However, in the case of ThS, since an organic-based washing method using ethanol is applied, an additional process of washing with a water-base solvent is required later. 3) Finally, ThS has little advantage for applying into clinical trial of AD diagnosis because it has been not proved in terms of safety as chemical probe. However, various biological benefits of fucoxanthin have been confirmed and reported in many studies, suggesting that fucoxanthin can be useful in clinical trials with high probability. We already described these points in the discussion part (Line 221-224 of Page 10). We deeply express our thanks for your insight comment.

Absolutely important would be the experiment where the authors would show labelling in the control WT (no AD) animal to see what fucoxanthin labels in a healthy brain.

Without these additional experiments I cannot agree with publication of the paper.

We deeply apologize our mistake for missing of control. As your suggestion, we did additional experiments using wild-type mice. We found that both Ab and tau were not detected by fucoxanthin in the wild-type mice brains. We added these results in the supplementary data.

References

  1. Oakley, H.; Cole, S. L.; Logan, S.; Maus, E.; Shao, P.; Craft, J.; Guillozet-Bongaarts, A.; Ohno, M.; Disterhoft, J.; Van Eldik, L.; Berry, R.; Vassar, R., Intraneuronal beta-amyloid aggregates, neurodegeneration, and neuron loss in transgenic mice with five familial Alzheimer's disease mutations: potential factors in amyloid plaque formation. J Neurosci. 2006, 26, (40), 10129-10140.
  2. Alzforum APPswe/PSEN1dE9 (C57BL6). Available online: https://www.alzforum.org/research-models/appswepsen1de9-c57bl6 (accessed on June 8, 2021),
  3. Alzforum 5xFAD (B6SJL). Available online: https://www.alzforum.org/research-models/5xfad-b6sjl (accessed on June 8, 2021),

Reviewer 3 Report

Authors performed an extensive study: "Validation of fucoxanthin from microalgae 2 Phaeodactylum tricornutum for the detection of amyloid burden in transgenic mouse models of Alzheimer’s disease".

Comparing fucoxanthin positive and Thioflavin S positive stained regions in the brains, authors determined that they are colocalized and that fucoxanthin can detect Aβ aggregates and serve as a new Aβ fluorescent imaging reagent in AD diagnosis.

Introduction is well written and, in my opinion, there is no need for changes.

Page 2, 2.1 Animals
Reference for the protocol from Jackson Laboratory is missing and should be added.

Page 2, 2.2 Preparation of fucoxanthin from P. tricornutum
Authors state that "an analysis grade solvent" was utilized for extraction, but they do not mention the solvent. In the next sentence they mention ethyl acetate, but the solvent name belongs to the sentence before.

Page 2, 2.3 Measurements of fucoxanthin fluorescent properties
"The emission scans were measured in 2 nm increments up to 850 nm in the range of 20 nm added to the absorption scan maximum by applying the excitation wavelength for fucoxanthin."
This statement is not clear to me and should be written in more reader friendly fashion.

Page 3, line 105,
There is a comma missing after 90.

Page 4, Figure 2
Very nice spectrum disappearance effect with decreasing concentration.
A peak at 686 nm in Fig. 2b should be highlighted.

Additionaly, Fig. 2c-f are not referenced in text and this should be corrected.

Page 5
On Fig. 2c there is noticeable lowering of intensity when you compare (20 uM 0 day and 20 uM 6 days). This is not present in other Fx concentrations. Is there any explanation for this?

Page 7
Author mention that "In this study, fucoxanthin from microalgae was extracted using ethanol", but in Materials they say something about ethyl acetate. Please correct if necessary.

Page 10
"But it takes at least 24-48 hours"
Please provide a reference.

Page 10
Lines 210-212 belong to Introduction.

Page 10
"Although fucoxanthin may not overcome all shortcomings of other dyes"
I think that shortcomings should be repeated here for a complete information.

Author Response

Reviewer #3:

First of all, we would like to express our genuine gratitude for the time and effort put into reviewing our manuscript and providing comments that make this even better. All authors have carefully reviewed the comments in detail and revised the manuscript with care.

Authors performed an extensive study: "Validation of fucoxanthin from microalgae 2 Phaeodactylum tricornutum for the detection of amyloid burden in transgenic mouse models of Alzheimer’s disease".

Comparing fucoxanthin positive and Thioflavin S positive stained regions in the brains, authors determined that they are colocalized and that fucoxanthin can detect Aβ aggregates and serve as a new Aβ fluorescent imaging reagent in AD diagnosis.

Introduction is well written and, in my opinion, there is no need for changes.

Page 2, 2.1 Animals
Reference for the protocol from Jackson Laboratory is missing and should be added.

Thank you for pointing out our mistakes. We added the protocol numbers provided by Jackson Laboratory as references for the protocol for genotyping of two transgenic mouse models (Line 64 of Page 2).

Page 2, 2.2 Preparation of fucoxanthin from P. tricornutum
Authors state that "an analysis grade solvent" was utilized for extraction, but they do not mention the solvent. In the next sentence they mention ethyl acetate, but the solvent name belongs to the sentence before.
Page 2, 2.3 Measurements of fucoxanthin fluorescent properties
"The emission scans were measured in 2 nm increments up to 850 nm in the range of 20 nm added to the absorption scan maximum by applying the excitation wavelength for fucoxanthin."
This statement is not clear to me and should be written in more reader friendly fashion.
Page 3, line 105,
There is a comma missing after 90.

We sincerely apologize for the inaccurate and ambiguous expressions. So, we have corrected each sentences containing the questionable content as follows:

  • “Briefly, freeze-dried P. tricornutum was supplied from Algaetech (Gangneung, Korea) and an analysis grade solvent (Daejeong) was utilised for fucoxanthin extraction. During extraction, ~”

→ Rephrased. “Freeze-dried P. tricornutum was supplied from Algaetech (Gangneung, Korea) and ethanol was purchased Daejung (Korea) for fucoxanthin extraction. Briefly, ~” (Line 71-73 of Page 2)

  • “The emission scans were measured in 2 nm increments up to 850 nm in the range of 20 nm added to the absorption scan maximum by applying the excitation wavelength for fucoxanthin.”

→ Rephrased. “By applying the excitation wavelength for fucoxanthin, the emission scans were measured in 2 nm increments up to 850 nm from 20 nm added absorption scan maximum.” (Line 82-85 of Page 2)

  • A comma missing after 90 was added in Page3 line 105.

Page 4, Figure 2
Very nice spectrum disappearance effect with decreasing concentration.
A peak at 686 nm in Fig. 2b should be highlighted.

Additionally, Fig. 2c-f are not referenced in text, and this should be corrected.

We appreciate that you have pointed out our fault and suggested improvements that could make the manuscript better. Regarding your comments, we modified Figure 2b to mark the peak at 686nm. Furthermore, Figure 2c-f was also added as a reference at appropriate location within the text (Line 134 of Page 5).

Page 5
On Fig. 2c there is noticeable lowering of intensity when you compare (20 uM 0 day and 20 uM 6 days). This is not present in other Fx concentrations. Is there any explanation for this?

We are grateful for your insightful comments. As you said, when Ab monomer (0 days) or oligomer (6 days) is added, a difference in fluorescence intensity is observed at a concentration of 2mM of fucoxanthin that does not appear in other concentrations. However, our purpose in conducting this experiment was to confirm that fucoxanthin does not lose its fluorescence when it reacts with Ab. In figure 2c, the two cases showed a difference in intensity, but we judged that this result was not a significant change and only interpreted that fucoxanthin had a fluorescence intensity that did not interfere with Ab detection. We think that further investigation chemically is needed to analyze the difference shown here. Thank you again for your in-deep comments.
Page 7
Author mention that "In this study, fucoxanthin from microalgae was extracted using ethanol", but in Materials they say something about ethyl acetate. Please correct if necessary.

Page 10
"But it takes at least 24-48 hours"
Please provide a reference.

Page 10
Lines 210-212 belong to Introduction.

We sincerely apologize for our mistakes that could provide misleading information for the readers by not being cautious with the description. First, in 2.2 Preparation of fucoxanthin from P. tricornutum, the misspelled ethyl acetate was corrected to ethanol. Next, "But it takes at least 24-48 hours" in the text was modified to "But it takes at least 16-24 hours", and we provided appropriate references. And finally, Page 10 Line 210-212 can be emphasized as an excellent advantage of fucoxanthin compared to ThS, so although it is described in detail in the introduction, it was mentioned once again in the discussion.

Page 10
"Although fucoxanthin may not overcome all shortcomings of other dyes"
I think that shortcomings should be repeated here for a complete information.

Thanks for the comments that allow our manuscript to be better. When I consider your response, support shortcomings about fucoxanthin are better to add in the manuscript. Thus, in discussion part where line 224-226, tau staining properties in 5XFAD by fucoxanthin content was added. 

Round 2

Reviewer 2 Report

/